# PARP Inhibitors in Pancreatic Cancer with Homologous Recombination Repair Gene Mutations: A Single-Institution Experience

**DOI:** 10.3390/cancers16203447

**Published:** 2024-10-11

**Authors:** Ruoyu Miao, Kirsten Blue, Katelyn Sommerer, Anand Shah, Sal Bottiglieri, Alex del Cueto, Darcy K. Berry, Teresa T. Ho, James Kevin Hicks, Dae Won Kim

**Affiliations:** 1Department of Gastrointestinal Oncology, Moffitt Cancer Center, Tampa, FL 33612, USA; rabbitmiao@hotmail.com (R.M.); kirsten.blue@moffitt.org (K.B.); katelyn.sommerer@moffitt.org (K.S.); anand.shah@moffitt.org (A.S.); salvatore.bottiglieri@moffitt.org (S.B.); 2Department of Pathology, Moffitt Cancer Center, Tampa, FL 33612, USA; adelcueto@carisls.com (A.d.C.); darcy.berry@moffitt.org (D.K.B.); teresa.ho@moffitt.org (T.T.H.); james.hicks@moffitt.org (J.K.H.)

**Keywords:** pancreatic cancer, homologous recombination repair, poly (ADP-ribose) polymerase inhibitor

## Abstract

**Simple Summary:**

The anticancer activity of PARP inhibitors in pancreatic cancer with mutations in HRR genes other than *BRCA* and *PALB2* remains inconclusive. The authors retrospectively reviewed the clinical characteristics and outcomes of patients with advanced pancreatic cancer harboring pathogenic germline and/or somatic HRR mutations who were treated with PARP inhibitors. PARP inhibitors may be considered for patients with advanced pancreatic cancer harboring pathogenic alterations of *BRCA* who cannot tolerate standard chemotherapy. Maintenance PARP inhibitor therapy may be considered in selected patients with non-*BRCA* and non-*PALB2* HRR mutations.

**Abstract:**

Background: Limited data are available regarding the anticancer activity of PARP inhibitors (PARPis) in pancreatic cancer with mutations in HRR genes other than *BRCA* and *PALB2*. Methods: We retrospectively reviewed the clinical characteristics and outcomes of 48 patients with advanced pancreatic cancer harboring pathogenic germline and/or somatic HRR mutations who were treated with PARPis. Results: Thirty patients had germline (g)HRR mutations only, twelve had somatic (s)HRR mutations only, and six had concomitant gHRR and sHRR mutations. The objective response rate (ORR) was 22%. The median progression-free survival (mPFS) and overall survival (mOS) were 6.9 and 11.5 months, respectively. Five patients received olaparib in the front-line setting due to borderline performance status. Their ORR was 20%, and their mPFS and mOS were both 11.3 months. The ORR was higher in patients with *BRCA* or *PALB2* mutations (germline or somatic) than in those with non-*BRCA*/*PALB2* mutations. Patients with somatic non-*BRCA*/*PALB2* variants had a shorter mPFS. Patients with concomitant gHRR/sHRR mutations or gHRR mutations alone had a significantly longer mPFS than those with sHRR mutations only. Conclusions: PARP inhibitors may be considered for patients with advanced pancreatic cancer harboring pathogenic alterations of *BRCA* who cannot tolerate standard chemotherapy. Maintenance PARPis can be considered in selected patients with non-*BRCA*/non-*PALB2* HRR mutations.

## 1. Introduction

Poly (ADP-ribose) polymerase (PARP) is an essential enzyme for the repair of DNA damage including strand breaks which frequently occur during cell proliferation. PARP inhibitors catalytically inhibit PARP and trap PARP on damaged DNA which subsequently stop replication forks from continuing. In normal cells, this would be fixed by homologous recombination (HR). However, in HR-deficient tumor cells, alternative error-prone DNA repair pathways may lead to DNA fragmentation and ultimately cell death [1,2,3,4]. Therefore, PARP inhibition is an attractive therapeutic approach in HR-deficient cancer. Approximately 20% of patients with pancreatic cancer have homologous recombination repair (HRR) gene mutations including *BRCA1*, *BRCA2*, *PALB2*, *ATM*, *BAP1*, *BARD1*, *BLM*, *BRIP1*, *CHEK2*, *FAM175A*, *FANCA*, *FANCC*, *NBN*, *RAD50*, *RAD51*, *RAD51C*, and *RTEL1* [5,6,7,8]. A PARP inhibitor, olaparib, has demonstrated significantly improved progression-free survival (PFS) in patients with metastatic castration-resistant prostate cancer harboring deleterious alterations of HRR genes including *BRCA1*, *BRCA2*, *ATM, BRIP1*, *BARD1*, *CDK12*, *CHEK1*, *CHEK2*, *FANCL*, *PALB2*, *PPP2R2A*, *RAD51B, RAD51C*, *RAD51D*, and *RAD54L* [9]. In metastatic pancreatic adenocarcinoma, maintenance olaparib prolonged PFS in patients with pathogenic alterations in germline *BRCA1 or BRCA2* compared to the placebo (7.4 months vs. 3.8 months) in the phase 3 POLO trial [10,11]. In the single-arm phase 2 trial of maintenance rucaparib in patients with pathogenic alterations in germline or somatic *BRCA1*, *BRCA2*, or *PALB2*, the median PFS was 13.1 months, and the response rate was 41% in patients with pathogenic alterations in germline *BRCA2*, 50% with germline *PALB2*, and 50% with germline *BRCA2* [12]. However, limited data are available regarding the anticancer activity of PARP inhibitors in other HRR-gene-mutated pancreatic cancers. In this study, we conducted a retrospective review of patients with HRR-gene-mutated pancreatic adenocarcinoma treated with PARP inhibitors at our institution.

## 2. Materials and Methods

### 2.1. Patient Selection and Clinical Data Collection

We retrospectively reviewed all patients with advanced pancreatic adenocarcinoma harboring pathogenic germline or somatic mutations in HRR genes who were treated with PARP inhibitors that started between 1 March 2018 and 31 July 2023 at Moffitt Cancer Center. Eligible patients were identified from pharmacies. The assays that were used to identify somatic HRR mutations included our in-house next-generation sequencing (NGS) assay (Moffitt Star) [13] and commercial NGS assays including FoundationOne^®^ CDx (Foundation Medicine, Cambridge, MA, USA), CARIS^®^ MI Tumor Seek Hybrid™ (Caris Life Sciences, Irving, TX, USA), FoundationOne^®^ Liquid CDx (liquid biopsy; Foundation Medicine, Cambridge, MA, USA), and Guardant360^®^ CDx (liquid biopsy; Guardant Health, Redwood City, CA, USA). Broad panel testing through Invitae^®^ (Invitae Corpo-ration, San Francisco, CA, USA), Myriad Genetics^®^ (Myriad Genetics, Salt Lake City, UT, USA), or Ambry Genetics^®^ (Ambry Genetics, Aliso Viejo, CA, USA) was used to identify germline HRR mutations. A comprehensive chart review was completed for each eligible patient utilizing the electronic health record. Patient demographics, primary tumor characteristics, and treatment and clinical outcomes were collected. The study cut-off date was 15 March 2024. This study was reviewed and approved by the Institutional Review Board at Moffitt Cancer Center.

### 2.2. Treatment Assessment

Treatment response was assessed according to Response Evaluation Criteria in Solid Tumors (RECISTs) 1.1. An objective response rate (ORR) was defined as the percentage of patients who achieved complete response (CR) or partial response (PR). Disease control rate (DCR) was defined as the percentage of patients who achieved either complete response (CR), partial response (PR), or stable disease (SD). Progression-free survival (PFS) was defined as the time interval from the initiation of a PARP inhibitor to disease progression or death. Overall survival (OS) was defined as the time interval from the initiation of a PARP inhibitor to death from any cause or last-contact date.

### 2.3. Statistical Analysis

Statistical significance between groups was analyzed using the chi-square test or Fisher’s exact test for categorical variables and Student’s *t*-test or the Mann–Whitney U non-parametric test for continuous variables. The estimated PFS and OS were derived using the Kaplan–Meier method and compared by the Mantel–Cox log-rank test. The statistical analyses were performed using IBM SPSS Statistics for Windows, version 28.0 (IBM Corp, Armonk, NY, USA). Kaplan–Meier survival curves were generated in R (version 4.2.1; www.r-project.org (accessed on 6 June 2024)) using the “survminer” package and the “ggsurvplot” function. All reported *p* values were two-sided. The level of significance was set at *p* < 0.05.

## 3. Results

### 3.1. Patient Characteristics

A total number of 48 patients were identified. Patient demographics and characteristics are summarized in Table 1. The majority were male (54%) with a median age of 71 years (range: 42–90 years). Forty-seven patients received olaparib and one was treated with rucaparib. A PARP inhibitor was given to 9 patients (19%) with locally advanced disease and 39 patients (81%) with distant metastases. Five patients (10%) received a PARP inhibitor in the front-line setting due to borderline performance status (PS). The remaining 43 patients (90%) received it as maintenance after chemotherapy. Twenty-five patients received platinum-based chemotherapy and eighteen patients received non-platinum-based chemotherapy prior to maintenance PARP inhibitor administration.

Thirty-six patients were found to have pathogenic germline mutations in HRR genes, including *BRCA2* (*n* = 11), *ATM* (*n* = 6), *BRCA1* (*n* = 3), *PALB2* (*n* = 3), *FANCA* (*n* = 3), *NBN* (*n* = 3), *CHEK2* (*n* = 2), *FANCM* (*n* = 2), and *RAD51C* (*n* = 2), and one each in *BARD1, BRIP1, FANCC, FANCG,* and *WRN* (Appendix A). Four patients had pathogenic germline mutations in two HRR genes (*ATM/FANCC*, *BRCA2/BRIP1*, *BRCA2/NBN*, and *RAD51C/FANCA*, respectively). Eighteen patients had pathogenic somatic mutations in HRR genes, including *ATM* (*n* = 7), *BRCA2* (*n* = 5), and *ARID1A* (*n* = 3), and one each in *BAP1*, *CHEK2*, *NBN*, and *PALB2*. One patient was found to have pathogenic somatic mutations in two genes (*ARID1A/NBN*). Of note, six patients had concomitant pathogenic germline (g) mutation and somatic (s) mutation, including g*ATM*/s*ATM*, g*BRCA2*/s*ATM*, g*NBN*/s*BRCA2*, g*RAD51C*/s*ARID1A*, g*PALB2*/s*BRCA2*, and g*PALB2*/s*PALB2*.

Of the 34 patients with NGS results, 22 (65%) had somatic mutations in *KRAS*, most commonly *KRAS G12V* (*n* = 9, 26%), followed by *KRAS G12D* (*n* = 6, 18%), *KRAS G12R* (*n* = 3, 9%), *KRAS Q61H* (*n* = 3, 9%), and *KRAS G12S* (*n* =1, 3%). One patient (3%) was found to have *NRAS G12D*. The remaining 11 patients (33%) were *RAS* wild type (Table 1). s*TP53* mutation was detected in 19 patients (56%). Nine patients (26%) had mutations in s*CDKN2A/B*.

### 3.2. Efficacy

Forty-six patients had follow-up scans that were evaluable for treatment response (Table 2). The overall ORR and DCR were 22% and 78%, respectively. Ten patients achieved partial response (PR, 22%), including two g*BRCA2*, one g*BRCA1*, one s*BRCA2*, one s*ATM*, one g*BRCA2*/g*BRIP1*, one g*BRCA2*/s*ATM*, one g*PALB2*/s*BRCA2*, one g*PALB2*/s*PALB2*, and one g*NBN*/s*BRCA2*. Stable disease (SD) was observed in twenty-six patients (57%), including five g*BRCA2*, three g*ATM*, three s*ATM*, two s*BRCA2*, two g*CHEK2*, two g*FANCM*, one g*PALB2*, one g*FANCA*, one g*FANCG*, one g*WRN*, one g*NBN*, one g*BRCA2*/g*NBN*, one g*RAD51C*/g*FANCA*, one g*RAD51C*/s*ARID1A*, and one g*ATM*/s*ATM*. Progressive disease (PD) was found in 10 patients (22%). The ORR was the highest in the patients with concomitant germline and somatic HRR mutations (67%). Among the 28 patients with only germline HRR mutations, the ORR was 14% (29% in those with g*BRCA* or g*PALB2* mutations and 0% in those with germline non-*BRCA* non-*PALB2* mutations). Among the 12 patients with only somatic HRR mutations, the ORR was 17% (33% in those with s*BRCA* or s*PALB2* mutations and 11% (1/9) in those with somatic non-*BRCA* non-*PALB2* mutations) (Table 2).

For all patients, the median PFS was 6.9 months (95% confidence interval (CI): 3.4–10.4) and the median OS was 11.5 months (95% CI: 6.0–17.0) (Table 3). Patients with concomitant germline and somatic HRR mutations or germline HRR mutations alone had significantly longer PFS compared to those with somatic HRR mutations only (median: 10.2 vs. 8.3 vs. 3.0 months; *p* = 0.025) (Table 3 and Figure 1A). OS was numerically longer in patients with concomitant germline and somatic HRR mutations and germline HRR mutations only but was not statistically significant (median: 11.5 vs. 14.0 vs. 6.8 months; *p* = 0.101) (Table 3 and Figure 1B). Among patients with germline or somatic HRR mutations only, median PFS and OS were 6.1 months and 8.6 months among those with g*BRCA* or g*PALB2* mutations, 8.3 months and 17.6 months with other germline HRR mutations, 12.2 months and 16.2 months with s*BRCA* or s*PALB2* mutations, and 2.6 months and 5.9 months with somatic non-*BRCA* non-*PALB2* mutations (Table 3, Figure 1C,D).

Among the 27 patients with non-*BRCA* and non-*PALB2* HRR mutations, 15 had stable disease for over four months after initiation of a PARP inhibitor. There were two patients with concomitant germline and somatic HRR mutations (g*ATM*/s*ATM*, g*RAD51C*/s*ARID1A*), one with two germline mutations (g*RAD51C*/g*FANCA*), nine with one germline mutation (three g*ATM*, two g*CHEK2*, and one each for g*WRN*, g*FANCA*, g*FANCM*, and g*NBN*), and three with one somatic mutation (all s*ATM*).

### 3.3. Maintenance after Chemotherapy

We identified patients who received non-platinum-based chemotherapy and were then switched to maintenance PARP inhibitors. We evaluated the efficacy of maintenance PARP inhibitors in this patient population.

PARP inhibitor therapy was given to 25 patients as maintenance following platinum-based chemotherapy and 18 patients following non-platinum-based chemotherapy. There were more patients with an Eastern Cooperative Oncology Group (ECOG) performance status (PS) score of 2, and fewer patients with an ECOG PS of 0 among those who received non-platinum-based chemotherapy (*p* = 0.003). There was no significant difference between maintenance after platinum-based and non-platinum-based chemotherapy in PFS (median: 5.1 vs. 10.2 months, *p* = 0.154) or OS (median: 11.5 vs. 14.0 months, *p* = 0.919) (Table 3). The ORR was 13% vs. 35% (*p* = 0.128) and the DCR was 67% vs. 88% (*p* = 0.152) (Table 4).

### 3.4. First Line for Borderline Performance Status

We identified five patients who received olaparib as first-line treatment, all with an ECOG PS of 2. One patient with g*BRCA2/*g*BRIP1* mutations achieved PR and four achieved stable disease with g*BRCA2* mutations (*n* = 2), a g*BRCA2/*g*NBN* mutation (*n* = 1), and an s*ATM* mutation (*n* = 1). The ORR was 20% in patients who received first-line therapy and 22% among those who received a PARP inhibitor as maintenance. The DCR was 100% and 76%, the median PFS was 11.3 and 6.9 months, and the median OS was 11.3 and 11.5 months, respectively (Table 3, Figure 1E,F).

### 3.5. Treatment after Refractory to a PARP Inhibitor

Six patients received platinum-based chemotherapy, and twenty patients received non-platinum-based chemotherapy after disease progression on a PARP inhibitor. The median PFS and OS from initiation of subsequent line chemotherapy were 3.6 months (95% CI: 0.0–7.3 months) and 7.8 months (0.0–16.9 months), respectively, for patients who received platinum-based chemotherapy and 2.6 months (0.0–5.3 months) and 5.7 months (1.1–10.4 months), respectively, for non-platinum-based chemotherapy. A circulating tumor DNA (ctDNA) test was evaluated in one patient after refractory to a PARP inhibitor but we did not observe any reversion mutations of HRR genes.

## 4. Discussion

In this retrospective analysis, patients with germline HRR mutations achieved longer PFS compared to those with somatic HRR mutations. The least benefit in PFS was observed among patients with somatic non-*BRCA*/*PALB2* mutations. Although not statistically significant, a similar trend was observed in OS. It has been reported that in metastatic breast cancer, patients with g*PALB2* or s*BRCA* mutations derived similar benefits from PARP inhibition as those with g*BRCA* alterations [14,15]. A meta-analysis demonstrated comparable ORR of PARP inhibitors in patients with various malignancy types harboring somatic vs. germline *BRCA* mutations [16]. In pancreatic cancer specifically, there was a numerically increased response rate in somatic *BRCA* patients (3/4, 75%) compared to germline *BRCA* patients (7/32, 21.9%), although not statistically significant (*p* = 0.12) [16]. However, data may be underrepresented as many existing clinical trials for PARP inhibitors did not specifically include patients with somatic *BRCA* alterations [10,11,16]. We did not find a significant difference in the frequency of *RAS*, *TP53*, or *CDKN2A* mutations between germline vs. somatic HRR mutations or *BRCA*/*PALB2* vs. non-*BRCA*/*PALB2* mutations. Interestingly, we observed higher wild-type *KRAS* in this cohort than historical data which showed around 10% wild-type *KRAS* in pancreatic adenocarcinoma [17,18,19]. This may suggest that HRR mutations could be one of the major oncogenic driver mutations in wild-type *KRAS* pancreatic cancer.

We observed significant clinical benefits of PARP inhibitor therapy in select patients with pathogenic non-*BRCA*/*PALB2* HRR mutations who achieved durable disease control. Most of these patients had germline HRR mutations, including g*ATM*, g*RAD51C,* g*FANCA*, g*CHEK2*, g*WRN*, g*FANCM*, and g*NBN*, whereas all three patients with somatic mutation alone had s*ATM* mutations. The frequency of *RAS*, *TP53*, or *CDKN2A* mutations was similar to that among the whole study cohort. Two parallel phase 2 nonrandomized clinical trials demonstrated a median PFS of 5.7 months and an OS of 13.6 months with olaparib monotherapy in patients with pretreated, advanced pancreatic cancer with DNA damage repair genetic alterations other than the germline *BRCA* variant, including g*ATM*, s*ATM*, g*PALB2*, s*ARID1A*, s*BRCA*, s*PTEN*, s*RAD51*, s*CCNE*, and s*FANCB*. Partial response was reported in one patient harboring an s*ATM* variant and one with a g*PALB2* variant [20]. This finding is similar to our study. We observed a median PFS of 6.9 months and an OS of 11.5 months as a maintenance setting in this study. However, another tumor-agnostic phase 2 study showed no significant anticancer activity of olaparib in somatic or germline *ATM*- or *CHEK2*-mutated cancers, although only four patients with pancreatic cancer were enrolled [21]. This discrepancy may be related to additional molecular and genetic alterations and aberrations not discovered with current tests. Further studies are warranted to identify additional biomarkers to predict responses to PARP inhibitor therapy in this population.

Acquired reversion mutations in *BRCA* or *PALB2* are uncommon but may restore the protein expression and potentially affect treatment outcomes including rapid progression on PARP inhibitors, resistance to subsequent platinum-based treatment, and poor overall survival [22,23,24,25,26]. In this study, only one patient was assessed for ctDNA after refractory to olaparib, but we did not observe any reversion mutations.

Several data points demonstrated clinical benefit of maintenance PARP inhibitor therapy [10,11,12,27,28,29]. However, the question remains as to whether the benefit during maintenance resulted from previous chemotherapy or purely from PARP inhibitor therapy, as maintenance requires at least stable disease from previous chemotherapy. In our study, objective responses were observed while on maintenance PARP inhibitors in several patients; therefore, the benefit can be inferred from PARP inhibitors.

Olaparib has been approved as a maintenance treatment for patients with g*BRCA*-mutated pancreatic cancer whose disease has not progressed on platinum-based chemotherapy. However, *BRCA* mutation status may not always be available before starting chemotherapy, and some patients cannot start platinum-based chemotherapy due to borderline or poor performance status. Currently, limited data are available for the efficacy of olaparib for this patient population. In our study, the maintenance olaparib after non-platinum-based chemotherapy and olaparib in the first-line setting for patients with borderline performance status showed comparable ORR, DCR, and survival benefit as maintenance treatment after platinum-based chemotherapy in *BRCA*-mutated pancreatic cancer, suggesting a potential benefit of olaparib in this patient population and clinical setting.

We have several limitations in this study. This is a retrospective study in a single institution with a limited number of patients. Very few patients with somatic *BRCA*/*PALB2*-mutated pancreatic cancer were enrolled in this study. In addition, patients in this study were selected based on different germline and/or somatic testing assays with different sensitivities, specificities, and number of targeted genes, and not all patients underwent both germline and somatic testing. Certain HRR mutations may be underestimated in our study. Further prospective studies with predefined germline and somatic testing assay(s) are needed to include other HRR alterations to verify our findings.

## 5. Conclusions

In summary, our study suggests that PARP inhibitors may be considered for patients with borderline performance status and advanced pancreatic cancer harboring pathogenic alterations of *BRCA* who cannot tolerate standard chemotherapy. Maintenance PARP inhibitors can also be considered even after non-platinum-based chemotherapy for patients with pathogenic *BRCA*-mutated pancreatic cancer. In addition, maintenance PARP inhibitor therapy may have clinical activity in selected patients with non-*BRCA* HRR-mutated pancreatic cancer. Further studies are needed to verify our findings, and further biomarker studies should be conducted to select patients with non-*BRCA* HRR-gene-mutated pancreatic cancer who may achieve clinical benefit from PARP inhibitors.

This article is a revised and expanded version of a paper entitled “PARP inhibitors in pancreatic cancer with homologous recombination repair gene mutations: A single-institution experience”, which was presented at the ESMO Gastrointestinal Cancers Congress 2024, in Munich, Germany, on 26–29 June 2024 [30].

## Figures and Tables

**Figure 1 cancers-16-03447-f001:**
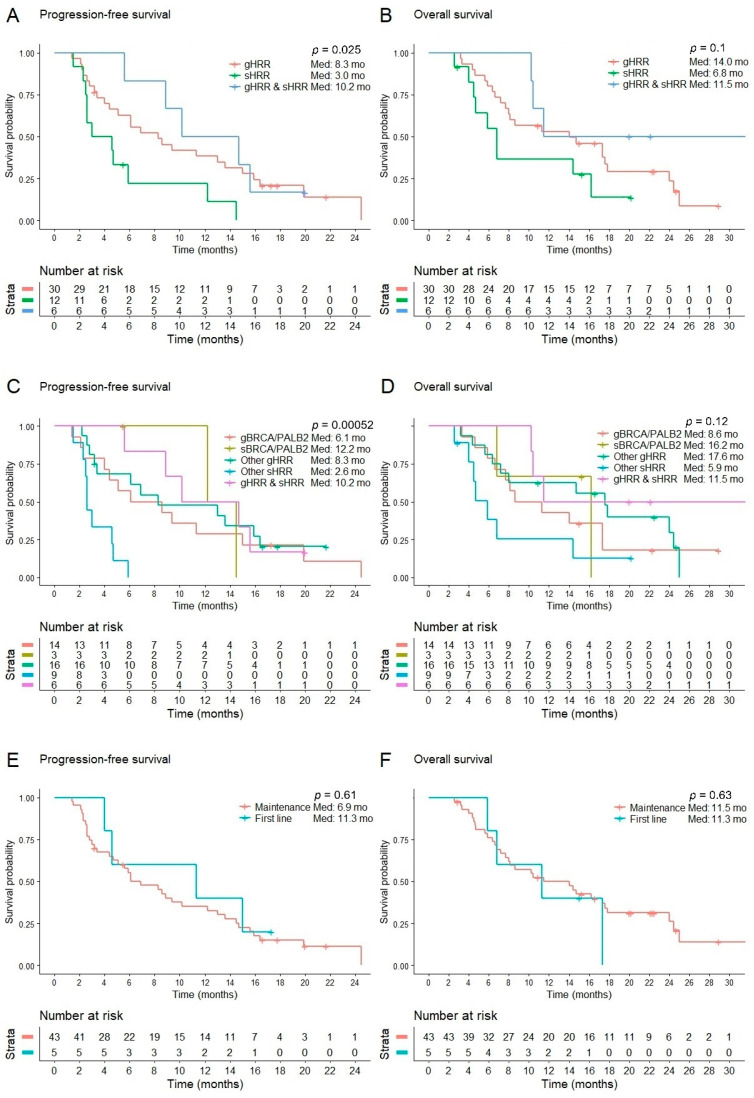
Kaplan–Meier curves of progression-free survival and overall survival stratified by germline vs. somatic mutation (**A**,**B**), HRR genes (**C**,**D**), and treatment setting (**E**,**F**).

**Table 1 cancers-16-03447-t001:** Patient characteristics.

Baseline Characteristics	Patients (*n* = 48)
Age, years, median (range)	71 (42–90)
Sex, *n* (%) Male Female	26 (54) 22 (46)
Race, *n* (%)	
White	48 (100)
ECOG at screening, *n* (%)	
0	7 (15)
1	28 (58)
2	13 (27)
Site of primary tumor, *n* (%)	
Head	23 (48)
Body	14 (29)
Tail	11 (23)
Staging, *n* (%)	
Locally advanced disease	9 (19)
Metastatic disease	39 (81)
Metastatic site, *n* (%)	*n* = 39
Liver	26 (67)
Lung	11 (28)
Peritoneum	8 (21)
Retroperitoneal lymph nodes	5 (13)
HRR mutation, *n* (%)	
Germline *BRCA* or *PALB2*	17 (35)
Somatic *BRCA* or *PALB2*	6 (13)
Germline non-*BRCA* non-*PALB2* HRR	19 (40)
Somatic non-*BRCA* non-*PALB2* HRR	12 (25)
RAS status, *n* (%)	
Wild type	11 (23)
*KRAS G12D*	6 (13)
*KRAS G12R*	3 (6)
*KRAS G12S*	1 (2)
*KRAS G12V*	9 (19)
*KRAS Q61H*	3 (6)
*NRAS G12D*	1 (2)
Unknown	14 (29)
Treatment setting, *n* (%)	
First-line monotherapy	5 (10)
First-line maintenance therapy	32 (74)
Subsequent line maintenance therapy	11 (26)
Previous chemotherapy, *n* (%)	*n* = 43
FOLFIRINOX	25 (58)
Gemcitabine/nab-paclitaxel	21 (49)
Gemcitabine/cisplatin	3 (7)
5FU/liposomal irinotecan	3 (7)
FOLFOX	2 (5)
Liposomal irinotecan monotherapy	1 (2)
Gemcitabine monotherapy	1 (2)
Other (trametinib, cobimetinib)	1 (2)

ECOG, Eastern Cooperative Oncology Group. HRR, homologous recombination repair.

**Table 2 cancers-16-03447-t002:** Best overall response by HRR gene mutation.

Best Overall Response	Total Evaluable	Germline *BRCA* or *PALB2* *	Somatic *BRCA* or *PALB2* *	Germline Non-*BRCA* Non-*PALB2* HRR *	Somatic Non-*BRCA* Non-*PALB2* HRR *	Germline and Somatic HRR ^#^	*p* Value
	*n* = 46	*n* = 14	*n* = 3	*n* = 14	*n* = 9	*n*= 6	
PR, *n* (%)	10 (22)	4 (29)	1 (33)		1 (11)	4 (67)	
SD, *n* (%)	26 (57)	7 (50)	2 (67)	12 (86)	3 (33)	2 (33)	
PD, *n* (%)	10 (22)	3 (21)		2 (14)	5 (56)		
ORR (CR + PR), *n* (%)	10 (22)	4 (29)	1 (33)	0 (0)	1 (11)	4 (67)	0.016
DCR (CR + PR + SD), *n* (%)	36 (78)	11 (79)	3 (100)	12 (86)	4 (44)	6 (100)	0.061

* Excluding patients with concomitant germline and somatic HRR mutations. ^#^ Patients with concomitant germline and somatic HRR mutations. CR, complete response. PR, partial response. SD, stable disease. PD, progressive disease. ORR, objective response rate. DCR, disease control rate.

**Table 3 cancers-16-03447-t003:** Survival estimates.

			Progression-Free Survival (Months)			Overall Survival (Months)		
		*n*	Median	95% CI	Log-Rank *p*	Median	95% CI	Log-Rank *p*
All		48	6.9	3.4–10.4		11.5	6.0–17.0	
HRR mutation								
	*BRCA*/*PALB2*	21	9.4	5.6–13.2	0.662	11.5	5.8–17.2	0.665
	Non-*BRCA*/*PALB2*	27	5.9	3.5–8.3		14.4	5.4–23.4	
	Germline mutation *	30	8.3	4.0–12.6	0.025	14.0	5.7–22.3	0.101
	Somatic mutation *	12	3.0	0.0–6.4		6.8	5.4–8.2	
	Concomitant germline and somatic ^#^	6	10.2	3.2–17.2		11.5	0.0–24.3	
	Germline *BRCA*/*PALB2* *	14	6.1	0.0–12.5	<0.001	8.6	2.7–14.5	0.121
	Somatic *BRCA*/*PALB2* *	3	12.2			16.2		
	Germline non-*BRCA*/*PALB2* *	16	8.3	0.0–16.8		17.6	12.6–22.6	
	Somatic non-*BRCA*/*PALB2* *	9	2.6	2.5–2.7		5.9	4.0–7.8	
	Concomitant germline and somatic ^#^	6	10.2	3.2–17.2		11.5	0.0–24.3	
Treatment setting								
	Maintenance	43	6.9	3.6–10.2	0.608	11.5	4.3–18.7	0.629
	First line	5	11.3	0.0–25.7		11.3	1.6–21.0	
Chemotherapy prior to maintenance								
	Platinum based	25	5.1	0.7–9.5	0.154	14.0	4.2–23.8	0.919
	Non-platinum based	18	10.2	2.5–17.9		11.5	3.5–19.5	

* Excluding patients with concomitant germline and somatic HRR mutations. ^#^ Patients with concomitant germline and somatic HRR mutations.

**Table 4 cancers-16-03447-t004:** Performance status and best overall response by treatment setting of PARP inhibitors.

	Total	Maintenance after Platinum	Maintenance after Non-Platinum	First Line
ECOG	*n* = 48	*n* = 25	*n* = 18	*n* = 5
0, *n* (%)	7 (15)	7 (28)	0 (0)	0 (0)
1, *n* (%)	28 (58)	17 (68)	11 (61)	0 (0)
2, *n* (%)	13 (27)	1 (4)	7 (39)	5 (100)
Best overall response	Evaluable *n* = 46	*n* = 24	*n* = 17	*n* = 5
PR, *n* (%)	10 (22)	3 (13)	6 (35)	1 (20)
SD, *n* (%)	26 (57)	13 (54)	9 (53)	4 (80)
PD, *n* (%)	10 (22)	8 (33)	2 (12)	
ORR (CR + PR), *n* (%)	10 (22)	3 (13)	6 (35)	1 (20)
DCR (CR + PR + SD), *n* (%)	36 (78)	16 (67)	15 (88)	5 (100)

ECOG, Eastern Cooperative Oncology Group. CR, complete response. PR, partial response. SD, stable disease. PD, progressive disease. ORR, objective response rate. DCR, disease control rate.

## Data Availability

The original contributions presented in this study are included in the article/Appendix A. Further inquiries can be directed to the corresponding author/s.

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
