# Peer review of "PARP Inhibitors in Pancreatic Cancer with Homologous Recombination Repair Gene Mutations: A Single-Institution Experience"

_cancers, 2024, doi:10.3390/cancers16203447_

Round 1
Reviewer 1 Report
Comments and Suggestions for Authors
This manuscript retrospectively analyzed the efficacy of the PARP inhibitor in pancreatic patients with mutations in homologous recombination genes. The data supports the authors' conclusion and will benefit the scientific community and patients.
To improve the quality of the manuscript, I would like to request that more information be added. The germline and somatic mutations in HR genes found in this report should be included as Supplemental Data. The publications that validated these mutations as deleterious must also be cited properly. Sometimes a mutation was mis-assigned as deleterious.
Other comment: The first paragraph of 3.2 Efficacy was the most difficult part to read in this manuscript. The authors should re-write this paragraph better. List the mutated genes in PR, SD, and PD. Then describe OPR so that readers will follow the data more easily, I think.
Example:
Forty-six patients had follow-up scans ……… [Table 2]. Overall ORR and DCR were 22% and 78%, respectively. Ten patients achieved partial response (PR, 22%), including two gBRCA2, one gBRCA1, ………..and one gNBN/sBRCA2. Stable disease (SD) was observed in 26 patients (57%), including # gBRCA2, # gBRCA1……. and xxx (please list all genes and combinations as you did in PR). Progressive disease (PD) was found in 10 patients (22%). OPR was the highest in the patients with concomitant germline and somatic HRR mutations (67%). Among the 28 patients with only germline HRR mutations, OPR was 14% (29% in those with ……..non-PALB2 mutations). Among the 12 patients with only somatic HRR mutations, ORP was 17% (33% in those …….non-PALB2 mutations)[Table 2].
Author Response
This manuscript retrospectively analyzed the efficacy of the PARP inhibitor in pancreatic patients with mutations in homologous recombination genes. The data supports the authors' conclusion and will benefit the scientific community and patients.
Comment 1: To improve the quality of the manuscript, I would like to request that more information be added. The germline and somatic mutations in HR genes found in this report should be included as Supplemental Data. The publications that validated these mutations as deleterious must also be cited properly. Sometimes a mutation was mis-assigned as deleterious.
Response 1: Thank you for pointing this out. We have updated the supplementary Table A1 (page 10, line 312, highlighted in red) with required information. We find it hard to cite the publications for all the mutations, as there would be over 100 references. Instead, we have searched these variants in the ClinVar database (https://www.ncbi.nlm.nih.gov/clinvar/) for the updated classification. There are three germline variants with conflicting classification (pathogenic/likely pathogenic vs. variant of uncertain significance) according to ClinVar, however, since there is no overwhelming evidence suggesting that the original classification of pathogenic/likely pathogenic variant was mis-assigned, we have kept these three cases. For somatic variants, there were 3 variants, two with premature stop codon and one with frameshift mutation, that were reported by the commercial assays as pathogenic/likely pathogenic but we were not able to find them in the ClinVar database.
Comment 2: The first paragraph of 3.2 Efficacy was the most difficult part to read in this manuscript. The authors should re-write this paragraph better. List the mutated genes in PR, SD, and PD. Then describe OPR so that readers will follow the data more easily, I think.
Example: Forty-six patients had follow-up scans ……… [Table 2]. Overall ORR and DCR were 22% and 78%, respectively. Ten patients achieved partial response (PR, 22%), including two gBRCA2, one gBRCA1, ………..and one gNBN/sBRCA2. Stable disease (SD) was observed in 26 patients (57%), including # gBRCA2, # gBRCA1……. and xxx (please list all genes and combinations as you did in PR). Progressive disease (PD) was found in 10 patients (22%). OPR was the highest in the patients with concomitant germline and somatic HRR mutations (67%). Among the 28 patients with only germline HRR mutations, OPR was 14% (29% in those with ……..non-PALB2 mutations). Among the 12 patients with only somatic HRR mutations, ORP was 17% (33% in those …….non-PALB2 mutations)[Table 2].
Response 2: Thank you for pointing this out. We have revised this paragraph (page 3, line 128, highlighted in red) according to the suggestion.
Reviewer 2 Report
Comments and Suggestions for Authors
Please , in the introduction, argue why have you joined in one category BRCA and PALB2 mutated patients.
In the methodology, it is not clear whether all tests were applied to all patients (I guess probably not). As tests may have different sensibilities, specificities, and aimed study targets, that should be cleared. I suggest including a patient vs study type table, as well as highlighting the similarities and differences among the different studies, since which are the mutations that the patients carry is on the base of this study. "The assays that were used to identify somatic HRR mutations included our in-house next generation sequencing (NGS) 68 assay,FoundationOne® CDx CARIS® MI Tumor Seek Hybrid™, or Guardant360® CDx 69 (liquid biopsy). A broad panel testing through Invitae® was used to identify germline 70 HRR mutations".
I have not been feeling well, but this study seemed too important to overlook it. I apologize, I may be missing something. This report is insufficient to reach firm conclusions, still is probably relevant, in view of the extremely difficult to deal-with pancreatic cancer. It may inspire mutlicentric studies in the same sense. Importantly, the authors are aware of limitations and honestly entitle the paper highlighting it is a limited, single center study.
I decided to send this review assumming there will be a second review occasion.
Author Response
Comment 1: Please, in the introduction, argue why have you joined in one category BRCA and PALB2 mutated patients.
Response 1: Thank you for pointing this out. We have revised the introduction (page 2, line 55, highlighted in red) according to the suggestion.
Comment 2: In the methodology, it is not clear whether all tests were applied to all patients (I guess probably not). As tests may have different sensibilities, specificities, and aimed study targets, that should be cleared. I suggest including a patient vs study type table, as well as highlighting the similarities and differences among the different studies, since which are the mutations that the patients carry is on the base of this study. "The assays that were used to identify somatic HRR mutations included our in-house next generation sequencing (NGS) 68 assay,FoundationOne® CDx CARIS® MI Tumor Seek Hybrid™, or Guardant360® CDx 69 (liquid biopsy). A broad panel testing through Invitae® was used to identify germline 70 HRR mutations".
Response 2: Thank you for pointing this out. This is a retrospective study in which we collected data based on available clinical information. In real-world scenario, clinicians could order any commercially available assay and treat patients based on relevant positive findings. We have updated the supplementary Table A1 (page 10, line 312, highlighted in red) with required information including the assays that were utilized. However, a comparison among assays would be beyond the scope of this retrospective study. We do acknowledge that gathering information from different assays with different sensitivities, specificities and targeted genes represents one of the limitations of our study and have added this into discussion (page 9, line 269, highlighted in red). We will need prospective studies with predefined testing assay(s) in the future.
Comment 3: I have not been feeling well, but this study seemed too important to overlook it. I apologize, I may be missing something. This report is insufficient to reach firm conclusions, still is probably relevant, in view of the extremely difficult to deal-with pancreatic cancer. It may inspire mutlicentric studies in the same sense. Importantly, the authors are aware of limitations and honestly entitle the paper highlighting it is a limited, single center study.
I decided to send this review assumming there will be a second review occasion.
Response 3: We greatly appreciate the comments and suggestions from the reviewer.
Reviewer 3 Report
Comments and Suggestions for Authors
Miao and colleagues a single-institution retrospective study on the use of PARP1 inhibitors in Pancreatic Cancer cases with germline and/or somatic mutations in Homologous Recombination repair (HRR) genes. The authors present clearly the significance of investigating the impact of mutations in non-BRCA/PALB2 HRR genes in PC cases. The data is presented clearly with good figures and tables. However, some points raised concerns that need to be addressed before the study is ready for publication.
In the Material and Methods section, 2.1 patient selection and clinical data collection, the authors need to describe in more details their in-house NGS, on how samples were prepared, and which platform was used. Alternatively, authors could cite a previous paper that describes it better. The information on which assay was used to identify the genes should be included in the Appendix A, Table A1.
If available, the authors should include any information on LOH for germline HRR mutated samples. If not available, this need to be disclosed in the discussion section.
The authors show the benefit of PARPi maintenance treatment, however, 26 patients (~50%) presented disease progression on PARPi treatment. Although the authors didn't find any reversions on HRR genes, is there any association between the HRR mutations and the reversion phenotype?
I understand the study has caveats regarding the single-institution and number of patients and appreciate the authors caution on their statements. However, I don't agree the data is strong to suggest that PARPi may be considered for first line therapy. The authors should tone down and rephrase their conclusion statement (line 264), simple summary (line 17) and abstract (line 33).
Overall, I believe this study has the potential to be published after the revisions above.
Author Response
Miao and colleagues a single-institution retrospective study on the use of PARP1 inhibitors in Pancreatic Cancer cases with germline and/or somatic mutations in Homologous Recombination repair (HRR) genes. The authors present clearly the significance of investigating the impact of mutations in non-BRCA/PALB2 HRR genes in PC cases. The data is presented clearly with good figures and tables. However, some points raised concerns that need to be addressed before the study is ready for publication.
Comment 1: In the Material and Methods section, 2.1 patient selection and clinical data collection, the authors need to describe in more details their in-house NGS, on how samples were prepared, and which platform was used. Alternatively, authors could cite a previous paper that describes it better. The information on which assay was used to identify the genes should be included in the Appendix A, Table A1.
Response 1: Thank you for pointing this out. We have added the reference (page 2, line 70, highlighted in red) for our in-house NGS and updated the supplementary Table A1 (page 10, line 312, highlighted in red) with required information.
Comment 2: If available, the authors should include any information on LOH for germline HRR mutated samples. If not available, this need to be disclosed in the discussion section.
Response 2: Thank you for pointing this out. We have updated the supplementary Table A1 (page 10, line 312, highlighted in red) with required information.
Comment 3: The authors show the benefit of PARPi maintenance treatment, however, 26 patients (~50%) presented disease progression on PARPi treatment. Although the authors didn't find any reversions on HRR genes, is there any association between the HRR mutations and the reversion phenotype?
Response 3: In this study, only one patient was assessed for ctDNA after refractory to olaparib, but we did not observe any reversion mutations. Therefore our data is insufficient for us to comment on the association between the HRR mutations and the reversion phenotype. From the literature, in one study in ovarian cancer, BRCA reversion mutations were detected in both BRCA1 (n = 4) and BRCA2 (n = 4), either germline (n = 5) or somatic (n = 3) in origin [1]. In a pan-cancer cohort, fewer patients had detectable reversions in BRCA1 (4.3%) versus BRCA2 (7.6%), although the difference was not statistically significant [2]. In a metastatic castration-resistant prostate cancer cohort, reversion rates were similar for BRCA2 and BRCA1, irrespective of germline or somatic status, but higher in samples with a high tumor DNA fraction [3]. Reversion mutation was also found in PALB2 gene in patients with pancreatic cancer who progressed on rucaparib [4].
References:
- Lin KK, Harrell MI, Oza AM, et al. BRCA Reversion Mutations in Circulating Tu-mor DNA Predict Primary and Acquired Resistance to the PARP Inhibitor Rucaparib in High-Grade Ovarian Carcinoma. Cancer Discov. 2019 Feb;9(2):210-19.
- Nakamura K, Hayashi H, Kawano R, et al. BRCA1/2 reversion mutations in a pan-cancer cohort. Cancer Sci. 2024 Feb;115(2):635-47.
- Loehr A, Hussain A, Patnaik A, et al. Emergence of BRCA Reversion Mutations in Patients with Metastatic Castration-resistant Prostate Cancer After Treatment with Rucaparib. Eur Urol. 2023 Mar;83(3):200-209.
- Brown TJ, Yablonovitch A, Till JE, et al. The Clinical Implications of Reversions in Patients with Advanced Pancreatic Cancer and Pathogenic Variants in BRCA1, BRCA2, or PALB2 after Progression on Rucaparib. Clin Cancer Res. 2023 Dec 15;29(24):5207-5216.
Comment 4: I understand the study has caveats regarding the single-institution and number of patients and appreciate the authors caution on their statements. However, I don't agree the data is strong to suggest that PARPi may be considered for first line therapy. The authors should tone down and rephrase their conclusion statement (line 264), simple summary (line 17) and abstract (line 33).
Response 4: Thank you for pointing this out. We have revised the statement accordingly (page 1, line 17; page 1, line 32; page 9, line 276, highlighted in red).
Overall, I believe this study has the potential to be published after the revisions above.
Reviewer 4 Report
Comments and Suggestions for Authors
The authors describe clinical characteristics and outcome of a subset of patients affected by pancreatic adenocarcinomas. All data were obtained retrospectively.
The authors reported 36 out of 48 (75%) pancreatic cancers with pathogenic variant of HRR genes including 30 germline and 6 somatic variants. This percentage of mutated cancers is very high and of should be discussed in relationship to data from literature. Probably this is a bias because it is not clear the selection of cases.
The performance of PARPi treatments and the statistical evaluation are dependent from genetic classification. In this paper no data are available about genetic testing on tumour tissues and on peripheral blood , on allelic variant frequency in case od somatic pathogenic mutations,
The genetic characteristics of the samples is very relevant in order to better verify the therapy performance .
In addition data on histological features should be also included.
Comments on the Quality of English LanguageI have no comment about English Language
Author Response
Comment 1: The authors reported 36 out of 48 (75%) pancreatic cancers with pathogenic variant of HRR genes including 30 germline and 6 somatic variants. This percentage of mutated cancers is very high and of should be discussed in relationship to data from literature. Probably this is a bias because it is not clear the selection of cases.
Response 1: This study is a retrospective review of all patients with advanced pancreatic adenocarcinoma harboring pathogenic germline or somatic mutations in HRR genes who were treated with PARP inhibitors, as mentioned in ‘2.1. Patient selection and clinical data collection’. Therefore, all patients in this study carry at least one pathogenic variant of HRR genes. This percentage does not reflect the actual prevalence of HRR mutations in the general pancreatic cancer population.
Comment 2: The performance of PARPi treatments and the statistical evaluation are dependent from genetic classification. In this paper no data are available about genetic testing on tumour tissues and on peripheral blood, on allelic variant frequency in case of somatic pathogenic mutations.
The genetic characteristics of the samples is very relevant in order to better verify the therapy performance.
Response 2: Thank you for pointing this out. We have updated the supplementary Table A1 (page 10, line 312, highlighted in red) with required information.
Comment 3: In addition data on histological features should be also included.
Response 3: Thank you for pointing this out. However, we find it not possible to provide more detailed information on histological features in the majority of patients in this study. As this study focuses on PARP inhibitors in patients with locally advanced or metastatic pancreatic adenocarcinoma who were not candidates for surgery, the majority of tissue diagnosis was based on a needle biopsy instead of surgical resection. There is only one patient who previously underwent surgery and adjuvant chemotherapy for resectable disease but later progressed with metastatic disease.